# Lymph Node Adiposity and Metabolic Dysfunction-Associated Steatotic Liver Disease

**DOI:** 10.3390/biomedicines13010080

**Published:** 2025-01-01

**Authors:** Jessica M. Rubino, Natalie Yanzi Ring, Krishna Patel, Xiaoqing Xia, Todd A. MacKenzie, Roberta M. diFlorio-Alexander

**Affiliations:** 1Radiology Department, Dartmouth Hitchcock Medical Center, Lebanon, NH 03756, USA; 2Geisel School of Medicine at Dartmouth, Hanover, NH 03755, USA; 3Hartford Healthcare, Midstate Radiology Associates, Hartford, CT 06103, USA; 4Department of Biomedical Data Science, Dartmouth College, HB 7261, Lebanon, NH 03756, USA

**Keywords:** non-alcoholic fatty liver disease, fatty liver, steatohepatitis, obesity, metabolic disorders, metabolic dysfunction, steatosis, inflammation, lymph node adiposity

## Abstract

**Objective**: Metabolic dysfunction-associated steatotic liver disease (MASLD), previously known as the most common chronic liver disease, is soon to be the leading indication for liver transplantation; however, the diagnosis may remain occult for decades. There is a need for biomarkers that identify patients at risk for MASLD and patients at risk for disease progression to optimize patient management and outcomes. Lymph node adiposity (LNA) is a novel marker of adiposity identified within axillary lymph nodes on screening mammography. Recent studies have demonstrated a correlation between LNA and cardiometabolic disease and cardiovascular disease risk. This study aimed to investigate the association between MASLD and LNA to evaluate the potential of mammographic LNA to serve as an imaging biomarker of MASLD. **Methods**: We identified women with pathology-proven MASLD who had a liver biopsy and a screening mammogram within 12 months of the liver biopsy. This resulted in a sample size of 161 women for final analysis that met the inclusion criteria. We evaluated lymph node adiposity through multiple measurements of the largest axillary lymph node visualized on mammography and correlated LNA with MASLD histology. Statistical analysis using univariable and multivariable logistic regression and odds ratios was performed using R version 4.1.0 (2021), the R Foundation for Statistical Computing Platform. **Results**: We found a significant association between MASLD and mammographic LNA, defined as lymph node (LN) length > 16 mm (*p* = 0.0004) that remained significant after adjusting for clinical factors, including body mass index (BMI). We additionally found a significant association between LNA and metabolic dysfunction-associated steatohepatitis (MASH), identified via liver biopsy (*p* = 0.0048). **Conclusions**: Mammographic lymph node adiposity may serve as a helpful imaging biomarker of MASLD in women who have an elevated risk for the development of MASH.

## 1. Introduction

Metabolic dysfunction-associated steatotic liver disease (MASLD), formerly known as non-acholic fatty liver disease (NAFLD), has become the most common cause of chronic liver disease with a prevalence of 25–30% worldwide. As obesity rates continue to increase, MASLD is the most rapidly growing indication for liver transplantation, projected to be the leading indication for liver transplantation in the United States in the next decade [1,2,3]. MASLD includes a spectrum of liver changes ranging from bland liver steatosis characterized by the accumulation of fat within hepatocytes, to metabolic dysfunction-associated steatohepatitis (MASH) characterized by inflammation and hepatocyte injury that can subsequently lead to fibrosis, cirrhosis, and ultimately end-stage liver disease [1,2,3,4]. The diagnosis of MASLD requires either an imaging diagnosis or a histologic diagnosis of liver steatosis, while MASH can only be diagnosed with liver biopsy and histologic assessment. As MASLD is often asymptomatic and requires clinical suspicion for diagnosis, it is often under-recognized and may remain undiagnosed, even in advanced stages [1,2,5,6,7,8]. There is currently no universal screening test for steatotic liver disease. Therefore, widely available non-invasive biomarkers that can predict MASLD are needed to identify patients who would benefit from diagnostic testing and management strategies that optimize patient outcomes [1,2,3]. Until recently, MASLD was managed with diet and lifestyle changes without available pharmacologic treatments. However, this year, the Food and Drug Administration approved resmetirom for the treatment of MASH, a thyroid hormone receptor B-agonist that decreases the lipotoxicity of steatohepatits. The availability of a medical treatment for MASH increases the need to improve awareness, risk assessment, and diagnosis of MASLD to ensure early intervention and limit disease progression [9]. The ability to predict MASLD is complicated by the diverse nature of steatotic liver disease prevalence, disease progression, differences in underlying genetics, the presence of co-morbidities, response to treatment, and variable distribution of adipose tissue throughout the body [3,6,10]. While obesity is a co-morbid condition that is almost always associated with MASLD, body mass index (BMI) is a crude predictor of adiposity, and measures of visceral and muscle adiposity are far better predictors of MASLD risk than BMI [3].

The new nomenclature for liver steatosis as a metabolic dysfunction-associated liver disease highlights the importance of metabolic disturbances as the underlying root cause of MASLD. The pathogenic drivers of MASLD are linked to inflammatory pathways and insulin resistance [1]. Metabolic dysfunction is strongly linked to ectopic fat deposition characterized by fat depots distributed within and around organs that are not physiologically designed for adipose storage [1,10,11,12]. Many studies have shown that ectopic fat is associated with adverse health outcomes due to dysregulated metabolism and chronic inflammation [1,10,11,12,13]. Ectopic fat deposition is well characterized in the liver as steatotic liver disease and was also identified in the visceral cavity, pancreas, heart, and muscle [1,11,12]. While it was shown that the distribution of ectopic adipose is a stronger predictor of MASLD and disease progression than body mass index (BMI), assessing ectopic fat deposition in the abdomen and skeletal muscle requires advanced imaging with CT or MRI. These studies are usually obtained only in patients who undergo advanced imaging for other clinical indications because the cost and ionizing radiation of these advanced imaging techniques preclude their use as a screening exam for MASLD. The identification of ectopic fat depots on a routine screening exam such as mammography would allow for the assessment of ectopic adiposity in a larger group of patients who do not have underlying clinical indications requiring advanced imaging and thereby increase the ability to screen for increased risk of MAFLD.

Lymph node adiposity (LNA) is a novel marker of ectopic adiposity within the immune system that can be readily evaluated on screening mammography recommended in the United States for all women ages 40–74. Axillary lymph nodes are well visualized in most breast imaging studies, and they demonstrate differences in size and architecture due to increased ectopic adipose deposition within the lymph node (LN) hilum, resulting in an increase in overall LN size due to fat expansion of the LN (Figure 1). Lymph nodes are secondary lymphoid organs of the immune–lymphatic system embedded within adipose tissue throughout the body. Studies have shown a significant association between BMI and enlarged axillary lymph nodes due to increased fat accumulation within the LN hilum, independent of age and breast density [14,15]. While LNA is associated with an increase in LN length and LN width, LN length is the largest metric of LN size with the highest sensitivity for adverse health outcomes and the best inter-rater agreement (Figure 1) [14]. Based on our studies, most normal axillary LNs measure in the range of 6–15 mm and may increase in size when enlarged by ectopic fat deposition, occasionally reaching a size of 50 mm [14,15,16,17,18,19,20]. Lymph node morphology allows for the definitive discrimination between increased size due to benign adipose deposition versus increased size due to reactive or malignant adenopathy. Unlike the assessment of other ectopic fat depots requiring advanced imaging, LNA is easily assessed on screening mammograms, an imaging exam that is widely utilized to screen for breast cancer among women above the age of 40 [14,15].

Metabolic co-morbidities are often associated with MASLD, including type 2 diabetes, dyslipidemia, and hypertension, and many studies have shown that cardiovascular disease (CVD) is the leading cause of mortality among patients with MASLD [1,7,10]. To our knowledge, we are the only team to investigate the association between LNA, cardiometabolic disease, and breast cancer outcomes [16,17,18,19]. We have recently shown that LNA is associated with a 10-year risk of CVD [17]. Our prior studies have additionally demonstrated a correlation between LNA, type 2 diabetes, and dyslipidemia [17,18]. In addition, we have shown a correlation between LNA and lymph node metastases among patients with breast cancer that is maintained when adjusted for clinical variables [19]. All of our studies evaluated LNA on mammography, and one study also evaluated LNA on breast MRI. We showed a strong agreement between mammographic and breast MRI measures of axillary lymph node adiposity, suggesting accuracy for assessment of LNA across imaging modalities, including advance imaging with MRI [19]. The agreement between MRI and mammographic measure of LNA allowed us to use a more cost-effective mammographic assessment of LNA for assessing MASLD risk. The strong association of cardiometabolic disease with both MASLD and LNA suggests that LNA may represent an opportunistic imaging biomarker of steatotic liver disease with the potential to identify women who require additional testing to confirm the diagnosis of MASLD and who may be at risk for progression to MASH.

Screening mammography is highly utilized in the United States, and in addition to screening for breast cancer, mammography can be used to opportunistically screen for other adverse health outcomes [20,21]. This was well documented by the early adoption of breast arterial calcification (BAC) reporting, an incidental finding on screening mammography that is now known to be associated with CVD risk and events [20]. LNA may also be a marker of cardiometabolic disease risk due to adiposity-associated pathways of cardiometabolic disease. Furthermore, the immune–lymphatic system plays a crucial role in lipid metabolism, and we have demonstrated that a decrease in LNA after bariatric surgery is associated with the resolution of dyslipidemia independent of patient age and weight loss [18]. As organelles of the immune system, lymph nodes are also responsible for immune signaling, inflammation, and immune cell profiles in the liver and other organ systems. Given the critical role of both lipid metabolism and immune function in the pathogenesis and progression of MASLD, ectopic adipose within the immune system may be linked to an increased risk of MASLD. The high prevalence of mammographic screening makes an association between axillary LNA and MASLD clinically relevant. Mammography would potentially provide a biomarker of MASLD risk in women without additional cost or testing. This study aimed to evaluate the association between MASLD and lymph node adiposity identified on screening mammograms obtained within one year of MASLD diagnosis by liver biopsy.

## 2. Materials and Methods

An institutional review board approved this retrospective study. This study was limited to patients with a screening mammogram for the identification of axillary LNA and, as such, was limited to women 40 years of age and older. Our pathology databank was queried for all liver biopsies in women 40 years of age and older between 15 February 2005 and 8 July 2016, yielding 1681 patients. Women who did not have a screening mammogram within 12 months of liver biopsy were excluded, resulting in 356 patients with both a liver biopsy and a screening mammogram. Eighty-seven patients were excluded due to clinical history of breast cancer or histology results demonstrating alternative etiology for liver disease, including drug induced hepatitis, viral hepatitis, and malignancy such as cholangiocarcinoma, hepatocellular carcinoma, and metastatic disease. Mammograms for the remaining 269 patients were reviewed by a radiologist with over twenty years of experience (RDA). Patients with non-visible axillary lymph nodes (defined as <80% of the lymph node visible on the mammogram) within either axilla or the medial-lateral oblique (MLO) view were also excluded, resulting in a total of 161 patients for analysis (Figure 2).

Data were collected from the electronic medical record (Epic Systems Corporation, Verona, WI, USA). These data included documented history of hypertension (HTN), hyperlipidemia (HLD), type 2 diabetes (T2DM), and body mass index (BMI). Obesity was defined as BMI ≥ 30, overweight as BMI < 30 and ≥25, and normal weight as BMI < 25 and ≥20.

Histologic Analysis: Tissue samples obtained by both core needle biopsies and surgical biopsies were included in the analysis. A pathologist independently evaluated tissue samples for histologic features of MASLD using the NAFLD activity score (NAS score) criteria. The NAS scoring system utilizes the histologic evaluation of the degree of steatosis, amount of lobular inflammation, extent of hepatocyte ballooning, and severity of fibrosis to diagnose and grade MASLD. Steatosis is defined as >5% hepatocytes infiltrated by fat, most commonly seen as intracytoplasmic macroscopic fat. Evaluation for lobular inflammation was made by the degree of infiltration of inflammatory cells. Hepatocyte injury was determined by the degree of hepatocyte ballooning. The NAS score is based on a total of 8 points, up to 3 points for steatosis and lobular inflammation, and up to 2 points for hepatocyte ballooning. A NAS score > 4 was considered diagnostic of MASH.

Image Analysis: Using Selenia and Dimensions Hologic units (Hologic Incorporated, Bedford, MA, USA), two-dimensional full-field digital mammograms were obtained, and images were reviewed on Barco 3-megapixel MDCG-3221 monitors (Kortrijk, Belgium) with Philips PACS v3.6 (Philips Healthcare, Best, The Netherlands). A breast radiologist (RDA) and senior radiology resident (KP), who were blinded to the patients’ clinical histories, reviewed each mammogram for the presence of visible lymph nodes. The single largest LN within either the right or left MLO view was chosen as the index node and was included for analysis if at least 80% of the LN was visible on the mammogram. The largest lymph node was selected as the index node for analysis based on prior work showing that the largest lymph node is correlated with obesity and adverse health outcomes [16]. Measurements obtained of the index lymph node included total LN length and width and hilar length and width. The LN cortex was defined as the peripheral radio-dense portion of the LN, and the LN hilum was measured as the radiolucent central area. Examples of lymph node and hilar measurements are demonstrated in Figure 3.

Statistical Analysis: We evaluated the association between MASLD (steatosis and steatohepatitis) and age, body mass index (BMI), hypertension (HTN), type 2 diabetes (T2DM), dyslipidemia (HLD), and LN dimensions (total LN length, total LN width, hilar length, and hilar width) using univariable and multivariable logistic regression. Odds ratios are reported for the association of each LN metric with steatosis (and steatohepatitis), controlling for age, BMI, HTN, T2DM, and HLD. We reported ROC curves and the area under the curve (C-statistic) for various models. Both radiologists measured all LNs, and the inter-rater reliability between the two readers was assessed using the Pearson correlation coefficient, which was a standard interpretation of the strength of association. The measurements reported in the manuscript are reader 1 (RDA) measurements. No patients with missing data were included in the analysis. Statistical analysis was performed with R version 4.1.0 (2021), the R Foundation for Statistical Computing Platform.

## 3. Results

### 3.1. Patient Characteristics and Image Analysis

The average patient age in this population was 60.3 (range 42–86 years), and the mean BMI was 37.1 (range 19.73–59.60). There was only one patient with a BMI < 20 who had a BMI of 19.7 and did not have evidence of steatosis on liver biopsy. Axillary lymph nodes were visible on the index mammogram (obtained within 12 months of the liver biopsy) in 60% of patients (161/269). The threshold for designating normal versus enlarged fatty lymph nodes was 16 mm; Table 1 demonstrates patient characteristics based on normal versus fat-enlarged LNs. The inter-rater agreement was good to excellent between readers with intraclass correlation of 97.2% for LN length, 95.7% for LN hilar length, 70.3% for LN width, and 89.7% for hilar width.

### 3.2. Correlation Between Lymph Node Metrics and MASLD

On univariable and multivariable logistic regression analyses, total LN length and width and hilar length and width were significantly correlated with steatosis (Table 2). Odds ratios indicate that total LN width and hilar width had the largest association with steatosis. Steatohepatitis was significantly associated with hilar length, with a trend towards an association with total LN length in univariate analysis. However, this association did not persist on multivariable analysis (Table 3).

ROC curves were generated to assess the utility of LNA for predicting steatosis and steatohepatitis. A total lymph node length value of 16 mm was 68% sensitive and 59% specific for steatosis. Areas under the receiver operating curve (AUC) for lymph node measurements alone ranged from 61.9% to 68.7% for predicting steatosis, as shown in Figure 4. The AUC of LN measurements for predicting steatohepatis was 53.5–61.8. When lymph node measurements were combined with other clinical risk factors, including age, BMI, HTN, T2DM, and HLD, the AUC for steatosis was 83.5%, and the AUC for steatohepatitis was 74.7%.

## 4. Discussion

MASLD affects 25–30% of adults globally and continues to increase at an alarming pace, yet many people with MASLD remain undiagnosed in part due to the lack of a widely available screening biomarker [1,2,3,4,5,6,7,8,9]. Approximately 20% of patients with steatosis develop MASH, which is more likely to progress to cirrhosis and end-stage liver disease. However, disease prevalence and progression are heterogeneous, making the identification of patients at risk difficult. While liver biopsy is the only reliable method for making a diagnosis of MASH, finding biomarkers that are associated with an increased risk of MASH could improve the identification of patients who would benefit from biopsy and increased surveillance. Further, understanding the mechanisms driving the association between MASLD and new biomarkers of disease could advance our understanding of disease progression and provide targets for future treatment approaches. Our study demonstrated a significant correlation between mammographic LNA and MASLD that was maintained when adjusting for clinical variables (OR 3.08 (CI 1.47, 7.35), *p*-value 0.006). It further showed a significant correlation between LNA and MASH in univariate regression analysis (*p*-value = 0.048). These findings support our innovative hypothesis that LNA identified via screening mammography may be opportunistically utilized as a biomarker of MASLD and also support our prior studies showing that LNA is associated with cardiometabolic disease [17,18,20].

There are several mechanisms by which excess lymph node adiposity may be linked to steatotic liver disease. MASLD is a complex disease with variable disease risk, manifestation, progression, and treatment response [1]. Despite the heterogeneity of MASLD, adipose tissue dysfunction is uniformly recognized as the basic feature of MASLD [1]. Adipose dysfunction is linked to adipose distribution. Unlike lipid metabolism in subcutaneous fat depots, lipid metabolism in ectopic adipose depots is dysfunctional and is associated with chronic inflammation [11,12,22]. Studies have shown a correlation between MASLD and visceral and muscle adiposity that are linked to adipose dysfunction and inflammation [11,12,22]. Therefore, the most likely mechanism contributing to the association between MASLD and LNA observed in our study is adipose dysfunction within enlarged fatty lymph nodes that impacts local and systemic inflammation. We recently identified an increase in the average size of fat-filled adipocytes (adipocyte hypertrophy) within benign enlarged hyper-adipose axillary sentinel lymph nodes compared to normal lymph nodes [23]. Adipocyte hypertrophy is the dominant histologic feature of ectopic adipose, resulting in reduced oxygen diffusion across enlarged lipid-laden cells that leads to hypoxia, inflammation, and dysregulated metabolism [24]. Increased fat within hypertrophic adipocytes activates inflammatory pathways and ultimately triggers insulin resistance, a common denominator for most cardiometabolic diseases, including MASLD [25]. Adipocyte hypertrophy observed in our prior study therefore supports dysregulated metabolism and inflammation as potential mechanisms accounting for the observed association between LNA and MASLD in our current study.

Another potential mechanism contributing to correlation between LNA and MASLD may be linked to the central role of the immune–lymphatic system in cholesterol transport. The function of the immune–lymphatic system is intricately connected to lipid metabolism via reverse cholesterol transport (RCT), a process in which dietary lipids are absorbed by the bowel lymphatics and transported through lymph nodes and lymphatic channels before entering the liver [26]. The structural changes in LNA may alter the transport of lipids in the lymphatic system as they are transported from the bowel to the liver due to lymph node hilar fat compression on the traversing lymphatic vessels. The mass effect of excess fat on traversing lymphatics could result in decreased egress of lipids from the lymph nodes to systemic circulation. The impact of LNA on RCT through the lymphatics could subsequently change the absorption of lipids within the liver, resulting in increased systemic and hepatic lipids. These alterations could be directly associated with an increased risk of MASLD development.

Finally, MASLD and LNA may be mechanistically linked via changes in immune function that result from LNA. Adipocyte hypertrophy and inflammation may adversely impact lymph node function, resulting in changes in innate immune signaling and inflammation, key features that are thought to play a major role in the progression of steatosis to steatohepatitis [27,28,29]. Organ dysfunction is a feature of ectopic adipose deposition as evidence by altered liver and muscle function with adipose infiltration. Lymph nodes infiltrated by excess adipose may similarly exhibit altered cortical functions that negatively impact host immune response. Recent studies have evaluated the immune system’s role in the complex pathogenesis of MASLD at the cellular and humoral levels [27,28,29,30]. However, we are the first study to examine the association between ectopic immune system adiposity and ectopic liver steatosis. Obesity is associated with immune dysfunction complicated by an increased risk of infections, increased risk of mortality due to infectious processes, and increased lymphedema due to impaired lymphatic transport [31,32,33]. With the discovery of adipokines, it has become increasingly clear that adipose tissue is metabolically active, and that ectopic adipose tissue is associated with metabolic dysregulation, leading to inflammation and organ dysfunction. This was well described in MASLD and other ectopic fat depots but remains relatively unexplored in the immune system [11,34].

Our work stems from the premise that ectopic fat within lymph nodes exhibits mechanisms of dysfunctional metabolism and inflammation that were identified in other ectopic fat depots [11,34]. Ectopic lymph node fat deposition results in adipose expansion of the central fatty hilum through which vessels and lymphatics traverse, and the structural changes in increased central LN fat may adversely impact immune and lymphatic function both locally and systemically. The results of our current study suggest that LNA identified via screening mammograms may represent an imaging biomarker of MASLD that could be incorporated into risk models to identify women who might benefit from increased testing and surveillance thereby addressing the current clinical gap of undiagnosed MASLD.

We demonstrated a significant association between mammographic LNA and MASLD diagnosed with liver biopsy. Specifically, we showed a significant association between FIN and histologic NAS activity score > 4, indicating MASH. Axillary lymph nodes are visualized in 75–80% of screening mammograms, and nearly 76% of eligible women in the United States undergo screening mammography [21]. The magnitude of mammography utilization makes LNA a significant opportunistic screening opportunity for identifying MASLD risk. The major strength of our study is the exploration of a novel concept that lymph node adiposity may represent an opportunistic imaging biomarker of MASLD. This hypothesis is founded on our prior studies in which we recently reported that lymph node adiposity is associated with cardiometabolic disease and 10-year risk of cardiovascular disease [16,17,18]. The novelty of our work is identifying and evaluating the significance of ectopic adipose within the immune system manifest by axillary lymph node fat enlargement on breast imaging exams. To our knowledge, we are the only team to explore this novel concept. Similar to the association between other ectopic fat depots and MASLD, our findings confirm that body fat distribution is a stronger predictor of MASLD than BMI. Our study is further strengthened by stringent criteria used for the diagnosis of MASLD via liver biopsy to confirm MASLD with a histologic diagnosis.

Our study is limited by several factors. First, this was a retrospective study that evaluated patients from a single institution, and as such, this may impact reproducibility. However, we found strong agreement between two readers, suggesting that this technique is reproducible in future studies. Second, our study is limited by a modest sample size. Our inclusion criteria ensured that all patients had a histologic diagnosis of MASLD; and while this limited our sample size to women with a liver biopsy and a screening mammogram within 12 months, it provided a definitive histologic diagnosis of MASLD. Third, LNA as a potential imaging biomarker of MASLD would only be clinically applicable to women who undergo screening mammography and therefore is not generalizable to males or younger patients. Despite the limitation of this mammographic approach for identifying the risk of MASLD in larger populations, it is interesting to note that higher rates of disease progression to MASH were reported among postmenopausal women compared to men thereby making this gender-specific imaging biomarker a potential benefit [35,36].

If reproduced in larger trials, LNA may be leveraged opportunistically from screening mammograms to serve as an imaging biomarker of MASLD and MASH. Future studies could confirm our findings with larger multi-site trials. Additional studies could further compare LNA to other ectopic fat depots, such as visceral adipose tissue and muscle adipose, to determine the prognostic benefit of LNA compared to other fat depots thereby revealing potential mechanisms accounting for this association. With the advent of AI for image analysis, LNA could be extracted from screening mammograms without additional cost, interpretation time, or testing. LNA identified on screening mammography could prompt targeted screening for MASLD and advanced liver disease in women who are otherwise asymptomatic and without significant risk factors. If LNA is validated as an imaging biomarker of MASLD, future studies could investigate the underlying mechanisms accounting for this association and potentially reveal therapeutic targets for future treatment trials.

## 5. Conclusions

Lymph node adiposity visualized on screening mammograms may serve as an imaging biomarker of MASLD and MASH among women ages 40–74. With recent FDA approval of the first available medication to treat MASH, the ability to identify women at risk for liver steatosis early in the course of the disease may prevent complications of advanced liver disease via frequent monitoring of MASLD, and early intervention with management and treatment strategies that limit progression to fibrosis and end-stage liver disease.

## Figures and Tables

**Figure 1 biomedicines-13-00080-f001:**
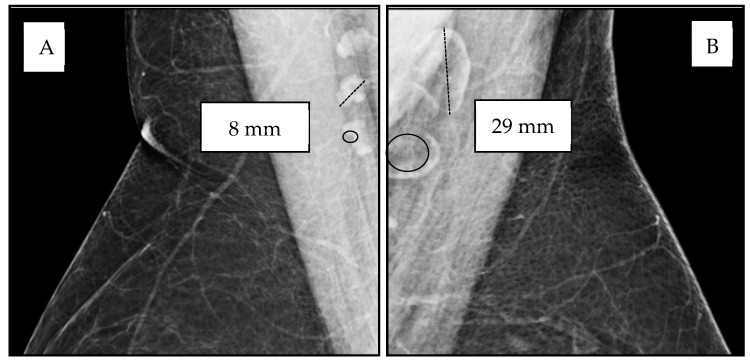
Variable lymph node size and morphology on mammographic medio-lateral oblique (MLO) views of the axilla. (**A**) Normal axillary lymph nodes measuring 8–11 mm (dotted line) with small physiologic “fatty notch” of lucent hilar fat (circle). (**B**) Fat-enlarged axillary lymph nodes measuring 25–29 mm (dotted line) due to increased lucent hilar fat (circle).

**Figure 2 biomedicines-13-00080-f002:**
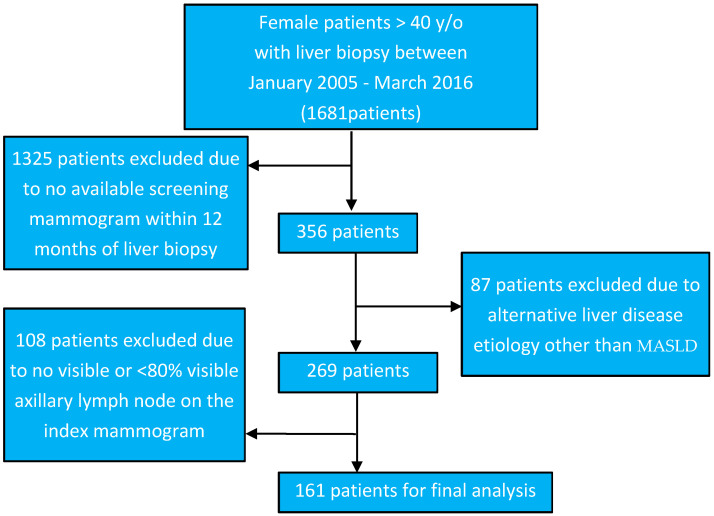
Data collection.

**Figure 3 biomedicines-13-00080-f003:**
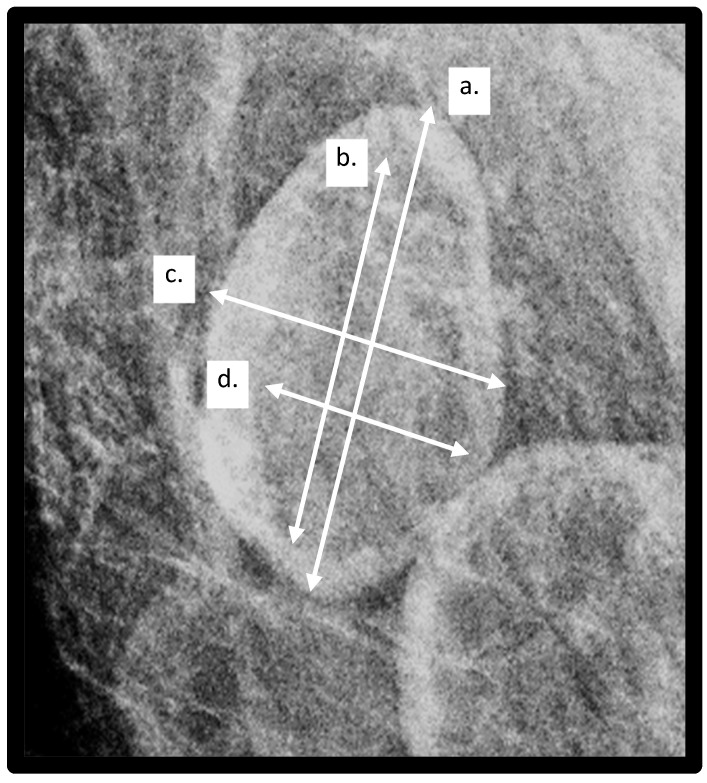
Lymph node (LN) measurements obtained: a—LN length, b—Hilar length, c—LN width, d—hilar width.

**Figure 4 biomedicines-13-00080-f004:**
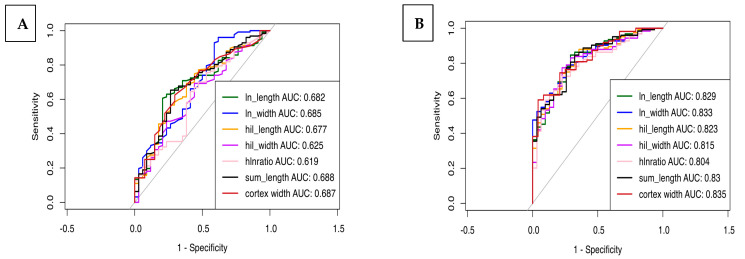
Performance of lymph node adiposity for predicting steatosis. (**A**) LN measurements alone yielded AUC of 61.9% to 68.7%. (**B**) LN measurements combined with clinical variables yielded AUC of 80.4% to 83.5%.

**Table 1 biomedicines-13-00080-t001:** Clinical characteristics of patients with and without lymph node adiposity (LNA) defined as LN length > 16 mm. BMI: body mass index; HTN: hypertension; T2DM: type 2 diabetes; HLD: hyperlipidemia. Steatosis and steatohepatitis based on histologic diagnosis.

Clinical Variable	Normal (LN Length < 16 mm) N = 60	LNA (LN Length > 16 mm) N = 101	*p* Value
Age (mean, SD)	60.6 (10.1)	59.8 (8.69)	0.61
BMI (mean, SD)	36.1 (8.72)	37.5 (7.77)	0.30
Steatosis	38 (63.3%)	89 (88.1%)	0.0004
Steatohepatitis (NAS > 4)	26 (43.3%)	68 (67.3%)	0.0048
HTN (N, %)	27 (38.3%)	56 (55.4%)	0.008
T2DM (N, %)	16 (26.7%)	39 (38.6%)	0.024
HLD (N, %)	23 (38.3%)	63 (62.4%)	0.017

**Table 2 biomedicines-13-00080-t002:** Association between steatosis and 10 mm increases in lymph node length using univariant and multivariant logistic regression.

	Univariable		Multivariable	
LN Metric	Odds Ratio (95% CI)	*p* Value	Odds Ratio (95% CI)	*p* Value
LN length	2.493 (1.419, 4.749)	0.0029	3.083(1.468, 7.351)	0.0060
LN width	7.656 (2.317, 30.661)	0.0019	9.646 (2.107, 57.571)	0.0069
Hilar length	2.552 (1.412, 5.029)	0.0037	2.688 (1.3, 6.250)	0.0134
Hilar width	3.246 (1.110, 10.570)	0.0394	3.878 (1.064, 16.070)	0.0486

**Table 3 biomedicines-13-00080-t003:** Association between steatohepatitis and 10 mm increases in lymph node length using univariate (*not univariant*) and multivariate (*not multivariant*) logistic regression.

	Univariable		Multivariable	
LN Metric	Odds Ratio (95% CI)	*p* Value	Odds Ratio (95% CI)	*p* Value
LN length	1.468 (0.998, 2.224)	0.0591	1.308 (0.849, 2.077)	0.2348
LN width	2.068 (0.933, 4.825)	0.081	1.551 (0.616, 4.083)	0.3593
Hilar length	1.548 (1.03, 2.411)	0.0428	1.339 (0.853, 2.179)	0.2177
Hilar width	1.846 (0.801, 4.414)	0.1569	1.746 (0.666, 4.741)	0.2628

## Data Availability

De-identified data are available upon request.

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
