# Peer review of "Lymph Node Adiposity and Metabolic Dysfunction-Associated Steatotic Liver Disease"

_biomedicines, 2025, doi:10.3390/biomedicines13010080_

Round 1
Reviewer 1 Report
Comments and Suggestions for Authors
General Comments:
The manuscript titled "Lymph node adiposity and Non-alcoholic Fatty Liver Disease " presents a compelling study which highlights that Lymph node adiposity visualized on screening mammograms may serve as an imaging biomarker of NAFLD and NASH among women, prompting additional testing and/or surveillance for women at risk for NAFLD and disease progression.
The topic is highly relevant; the experimental design is comprehensive. However, there are several areas in the manuscript that require substantial revisions before the manuscript can be considered for publication.
1. Abstract
The study presents a well-defined study objective that addresses a relevant issue in the field of non-alcoholic fatty liver disease (NAFLD). The investigation into using lymph node adiposity (LNA) as an imaging biomarker is a novel approach that could offer non-invasive diagnostic insights. The study is well-structured, with clear delineations of the objective, methods, results, and conclusion.
Strengths
Clarity and Focus: The study objective is clearly stated, and the research question is well-defined. The potential use of LNA as a biomarker for NAFLD is an innovative approach that addresses an unmet need in early diagnosis.
Novelty: The exploration of LNA as an imaging biomarker is intriguing and relevant. It adds a novel angle to the diagnosis of NAFLD, which is significant given the disease’s asymptomatic nature.
2. Methodological Design:
The methodology is straightforward, and the choice of correlating mammographic LNA with NAFLD histology is appropriate for the research question. The methods are generally well-described, but there is a lack of detail in certain areas. For instance, the exact protocols for Imaging Analysis and Histologic Analysis are not clearly specified. This information is crucial for reproducibility. Consider providing additional detail on the statistical methods used for adjustment and analysis. This would improve transparency and allow readers to better assess the validity of the results.
3. Introduction:
The introduction provides a good overview of the background, but the authors should justify the novelty of their proposed work. There are several literatures available which stated almost the same thing. The study focuses solely on women, as indicated in the methods. It might be beneficial to clarify the reasoning behind this gender-specific focus earlier in the abstract, particularly in the objective or methods. Suggest explicitly noting the exclusion of other genders if that was a deliberate methodological choice.
Terminology Consistency:
The term “lymph node adiposity” is used without a concise definition. A brief definition could enhance clarity, especially for readers who may not be familiar with LNA. Consider adding a short explanation in the introduction.
4. Results Presentation:
The results are statistically significant and clearly reported, indicating a strong association between LNA and both NAFLD and NASH, even after adjusting for BMI.
The resolution of figure 3 could have been better.
5. The literature review lacks depth and relies on outdated sources. It references some relevant studies but overlooks other key research in the field. Please verify the timeliness of your citations, prioritizing recent developments that could enhance your analysis. Ensure that all sources are up-to-date and pertinent. Incorporating newer studies will help to place your findings in a broader and more current research context.
6. Study Limitations:
The study could briefly mention potential limitations or confounding factors that might affect the association between LNA and NAFLD, such as age or other comorbid conditions.
7. How does the author determine the success of model establishment? The author should have clear indicators of successful model establishment. The manuscript seems to be incomplete and it could have been much better with more information and proper proposed molecular mechanistic pathways with a suitable figure for the proposed mechanisms of action of Lymph node adiposity and Non-alcoholic Fatty Liver Disease.
8. The conclusion succinctly summarizes the findings but could be expanded to more explicitly state the potential clinical implications of your research and suggest specific areas for future studies. It will be better if this section is carefully revised. While the conclusion suggests LNA could serve as a biomarker, a paragraph elaborating on the broader clinical implications would strengthen the abstract. For instance, how might this change current diagnostic protocols or what implications does it hold for routine mammographic screening?
9. Strengths and limitations: It will be better if you specifically write them in your manuscript.
10. There are several grammatical errors and awkward phrasings throughout the manuscript. I suggest a thorough revision of the manuscript for language issues, perhaps with the help of a professional editor.
Comments on the Quality of English LanguageThere are several grammatical errors and awkward phrasings throughout the manuscript. I suggest a thorough revision of the manuscript for language issues, perhaps with the help of a professional editor.
Reviewer 2 Report
Comments and Suggestions for Authors
The article „Lymph node adiposity and Non-alcoholic Fatty Liver Disease“ by Jessica Rubino et al. explores the potential of mammographic lymph node adiposity (LNA) as an imaging biomarker for diagnosing non-alcoholic fatty liver disease (NAFLD) and predicting its progression to non-alcoholic steatohepatitis (NASH). However, the motivation for the study is not convincing, as the nomenclature and the diagnosis criteria were updated.
Majors:
· The Template used for the manuscript is old, please consider using the new version.
· There is room in the Abstract for improvements. For example: the methods section in the abstract lacks specificity about sample size and statistical methods.
· I would recommend using the new nomenclature for fatty liver “Metabolic dysfunction–associated steatotic liver disease (MASLD)”. This aligns the study with current guidelines and ensures its relevance to contemporary research and clinical practice. Also, please consider elaborating on how LNA might contribute to the diagnostic criteria of MASLD.
· What are the limitations of existing diagnostic tools like MRI/CT and the advantages of mammography (cost-effectiveness and availability)?
· 108 Patients were excluded due to non-visible axillary lymph. Was this due to technical limitations, patient anatomy, or other factors? Could this exclusion have introduced selection bias?
· In the data collection BMI < 25 was defined as normal weight. Are there patients with a BMI < 20, which is considered underweight?
· The description of LN measurements is clear but would benefit from additional context. For example, explain why the largest visible LN was chosen and how this choice reflects ectopic adiposity trends. Also, are there any other factors that could influence the size of LN?
· The threshold of 16 mm for normal versus enlarged lymph nodes is mentioned but not justified. Provide references or prior studies that support this cutoff value, as it directly impacts the interpretation of results.
· While the intraclass correlation coefficients are reported, the lower reliability for LN width (70.3%) raises concerns. Consider discussing why this metric showed reduced agreement and its implications for result accuracy.
· While LN width and hilar width show strong correlations with steatosis, the associations with steatohepatitis are less robust. Discuss why LN metrics are less predictive for steatohepatitis and whether other clinical or histological features could enhance predictive models.
· Which clinical covariates had the largest impact on diminishing these associations?
· The study focuses exclusively on women aged ≥ 40. Briefly address whether findings can be generalized to other populations (e.g., younger women or males) and the implications of this limitation.
Minors:
· The third affiliation is in a different font size.
· BMI is introduced in the abstract without prior definition.
· Introduction (Line 34) Non-acholic is missing an “l”.
· In Figure 1 the last part of the figure legend is missing.
· The flowchart is helpful but could be improved by clearly labeling the exclusion steps with specific reasons (e.g., “87 patients excluded due to alternate liver disease etiology such as viral hepatitis”).
· Mention how missing or incomplete data (if any) were handled in the statistical analysis.
· Table 1 is helpful, but the definition of variables like “steatosis” and “steatohepatitis (NAS > 4)” should be explicitly stated in the table or caption for standalone clarity.
· Table legend should be above the table, please consider correcting this for table 2 and table 3.
· Table 3 and the line numbers are overlapping.
· Please revise the Author Contributions part and delete the unnecessary parts e.g. “For research articles with several authors, a short paragraph specifying their individual contributions must be provided. The following statements should be used“. Please complete who did what
